# Investigating the IgM and IgG B Cell Receptor Repertoires and Expression of Ultralong Complementarity Determining Region 3 in Colostrum and Blood from Holstein-Friesian Cows at Calving

**DOI:** 10.3390/ani14192841

**Published:** 2024-10-02

**Authors:** Tess E. Altvater-Hughes, Harold P. Hodgins, Douglas C. Hodgins, Cathy A. Bauman, Marlene A. Paibomesai, Bonnie A. Mallard

**Affiliations:** 1Department of Pathobiology, Ontario Veterinary College, University of Guelph, Guelph, ON N1G 2W1, Canada; altvatet@uoguelph.ca (T.E.A.-H.); dhodgins@uoguelph.ca (D.C.H.); 2Department of Biology, University of Waterloo, Waterloo, ON N2L 3G1, Canada; hhodgins@uwaterloo.ca; 3Department of Population Medicine, Ontario Veterinary College, University of Guelph, Guelph, ON N1G 2W1, Canada; cbauman@uoguelph.ca; 4Ontario Ministry of Agriculture, Food and Agribusiness, Guelph, ON N1G 4Y2, Canada; marlene.paibomesai@ontario.ca

**Keywords:** colostrum, B cells, bovine, ultralong complementarity determining region 3

## Abstract

**Simple Summary:**

Newborn calves rely on consuming the first milk, colostrum, to transfer important proteins and cells for protection from pathogens. Lymphocytes compose a fraction of the cells transferred through the colostrum to the calf, known as T and B cells. In cattle, a subset of B cells possess a unique receptor with an ultralong protein insert, known as ultralong complementarity determining region 3 (CDR3), which may be crucial in neutralizing viruses. This study found that colostrum had a significantly higher percentage of B cells with ultralong CDR3s than B cells from the blood. This finding may indicate an important role and selective transfer of B cells in colostrum with an ultralong CDR3.

**Abstract:**

In cattle, colostral maternal immunoglobulins and lymphocytes transfer across the neonate’s intestinal epithelium to provide protection against pathogens. This study aimed to compare repertoires of B cell populations in blood and colostrum in cows for the first time, with an emphasis on ultralong complementarity determining region 3 (CDR3, ≥40 amino acids). Blood mononuclear cells (BMCs, *n*= 7) and colostral cells (*n* = 7) were isolated from Holstein-Friesian dairy cows. Magnetic-activated cell sorting was used to capture IgM and IgG B cells from BMCs. Colostral cells were harvested by centrifugation. RNA was extracted and cDNA was produced; IgM and IgG transcripts were amplified using polymerase chain reactions. Amplicons were sequenced using the Nanopore Native barcoding kit 24 V14 and MinION with R10.4 flow cells. In colostrum, there was a significantly greater percentage of IgM B cells with ultralong CDR3s (8.09% ± 1.73 standard error of the mean) compared to blood (4.22% ± 0.70, *p* = 0.05). There was a significantly greater percentage of IgG B cells in colostrum with ultralong CDR3s (12.98% ± 1.98) compared to blood (6.61% ± 1.11, *p* = 0.05). A higher percentage of IgM and IgG B cells with ultralong CDR3s in colostrum may be indicative of a potential role in protecting the neonate.

## 1. Introduction

Colostrum is crucial to the immunological protection of neonatal calves due to the lack of placental transfer of maternal factors, including immunoglobulins (Ig) [1]. Components including Ig, growth factors, and cells are transferred via colostrum to the gut and circulation of the neonatal calf, providing a rich and concentrated source of passive immunity [1,2]. To date, most research on bovine colostrum has focused heavily on acellular components and much less on the cellular fraction, especially B cells.

Colostrogenesis is the process of colostrum production in the mammary gland. The production of colostrum is considered to begin as early as 5 weeks before calving [3]. However, there is a considerable transfer of IgG from the blood to the mammary gland in the final days before calving [4,5]. Lymphocytes are present locally in the mammary gland as resident cells. Lymphocytes can also infiltrate the mammary gland through diapedesis but the kinetics of this during colostrogenesis is uncertain [6,7]. During lactation, the leukocyte populations within an uninfected mammary gland epithelial cell lining are mostly macrophages and lymphocytes [6]. It has been estimated that colostral secretions contain a cellular population of approximately 5 × 10^5^ cells/mL, typically composed of macrophages (21% to 46%) and neutrophils (37% to 40%), with a smaller percentage of lymphocytes (17% to 23%) [8]. However, most lymphocytes in colostrum are T cells (88%), with a smaller percentage of B cells (3.5%), and the remaining percentages belong to subsets, such as natural killer cells [8]. In the blood, B cells compose approximately 54.5% (± 4.5%) of circulating lymphocytes in multiparous cows [9].

Previous studies have shown colostral maternal cells (from the dam) can be taken up intact across the gut epithelium by her calf. There is increasing interest in whether these cells remain functional and contribute to immune response development and/or protection against pathogens [10,11]. Several groups have studied cell populations after feeding whole versus cell-free colostrum. Langel et al. (2015) [2] have reported that calves fed whole colostrum, containing live maternal lymphocytes, had a greater proportion of CD4+ T cells in blood at 1 day of age. Reber et al. [12,13] have reported that calves fed whole colostrum had differential expression of cell surface markers on monocytes and lymphocytes, suggesting an influence of maternal cells on cell activation and pathogen clearance [10]. In another study, calves fed colostral maternal cells and challenged with *Escherichia coli* had greater specific IgA and IgM antibody responses and shed fewer bacteria than calves fed colostrum without maternal cells [14]. Freezing or heat treatment results in the death of colostral cells [1], which may negatively impact colostral quality. Calves exhibit tolerance to maternally derived colostral cells from their own dams but colostral cells from unrelated cows are rejected [10].

Antibodies and B cell receptors (BCRs) are composed of two heavy chains and two light chains. Each chain contains a variable domain with four conserved framework regions (FWRs) with three hypervariable complementarity determining regions (CDRs) interspersed among them. These regions are coded by the variable (V), diversity (D), and joining (J) genes. Each recombination event produces a unique arrangement of VDJ genes, which are further diversified via junctional nucleotide additions/deletions and activation-induced cytidine deaminase (AID) random nucleotide additions. Each unique receptor is specific to a B cell clone, which, upon activation, can proliferate, leading to a clonal expansion event. The daughter clones will have the same receptor as the original single B cell clone.

In humans, typical CDR3s range in length from 3 to 20 aa with an unimodal distribution [15]. In cattle, however, the range is more extensive [16], with cattle having a trimodal distribution of CDR3 lengths [17]. There is a subset with short CDR3s (≤10 aa), medium CDR3s (>10 aa to <40 aa), and ultralong CDR3s (≥40 aa) [17]. The phenomenon of ultralong CDR3s has been identified in the *Bos* and *Bison* genera [18] but not in sheep [19], horses [20], or pigs [21]. The ultralong CDR3 protrudes from the antibody structure as an ascending ß-sheet stalk, followed by a folded knob domain, and then a descending stalk structure [22,23]. The structure is further diversified through cysteine–cysteine disulfide bonds within the knob region, which tend to appear in even numbers and generate unique folding patterns [16,24]. Light chain pairing can also influence the positioning of the protruding ultralong CDR3 domain [25]. The knob domain has been suggested to be solely responsible for antigen binding and, since it is protruding from the structure, can reach hidden and concave epitopes that are not otherwise easily accessible [23,26]. The functional importance of antibodies with ultralong CDR3s in the *Bos* and *Bison* genera is unknown; however, they may compensate for the limited VDJ genes available for recombination [27,28]. The structural diversity of antibodies with ultralong CDR3s is expected to play important roles in defense against pathogens [28].

Sok et al. (2017) [29] have investigated the neutralization abilities of the bovine ultralong CDR3s against human-specific viruses, such as human immunodeficiency virus (HIV). They reported broad neutralization by specific B cell clones with ultralong CDR3s against clades of HIV following the immunization of cattle with a recombinant HIV gp140 trimer [29]. The engineering of antibodies mimicking the structure and neutralization capabilities of bovine ultralong CDR3 antibodies is of current interest for applications in human medicine [30] and may hold similar benefits for veterinary species. The presence of B cells with ultralong CDR3s in bovine colostrum has not been previously reported. This manuscript contributes to the minimal literature available on bovine colostral B cells and offers the first study of the BCR repertoire of B cells in bovine colostrum.

Given the presence of ultralong CDR3s in B cells in bovine blood, it was hypothesized that B cells found in colostral secretions would include a subset expressing ultralong CDR3s, and that they would be found in greater proportions compared to blood, since colostral components tend to be highly concentrated to maximize health benefits to the neonate. The objectives of this study were to assess IgM and IgG BCR repertoires and investigate the incidence of ultralong CDR3s in colostrum- compared to blood-derived B cells.

## 2. Materials and Methods

### 2.1. Animals and Sampling

The Animal Care Committee of the University of Guelph approved all animal use in this study under Animal Use Protocol #4449. Animals were housed at the Ontario Dairy Research Centre near Elora, Ontario. Fifty mL of colostrum was collected from each of the 7 primiparous (*n* = 4) and multiparous (*n* = 3) Holstein-Friesian cows within 2 h of calving. Colostrum was milked out from each quarter, mixed well in a bucket milker, and collected using a ladle. The sample was placed on ice, transported to the laboratory, and processed immediately. At the time of calving, 20 mL of whole blood was collected in K2EDTA-coated blood tubes (Becton Dickinson, Franklin Lakes, NJ, USA, catalog# 02-657-32) from the same primiparous and multiparous cows.

### 2.2. Isolation of Blood Mononuclear Cells (BMCs) and Colostral Cells

In colostrum, a smaller percentage of lymphocytes are B cells (3.5%) [8] while, in blood, B cells compose approximately 54.5% (± 4.5%) [9]. First, 25 mL of fresh colostrum was diluted with 25 mL of phosphate-buffered saline (PBS) containing 0.5M EDTA. PBS used in subsequent steps did not contain EDTA. Samples were centrifuged at 700× *g* for 20 min at 4 °C. The fat from the colostrum was removed, the supernatant was discarded, and the pellet was resuspended in 10 mL of PBS. The sample was then centrifuged at 400× *g* for 10 min at room temperature (RT). The supernatant was discarded and the previous step was repeated to wash the pellet. Finally, the supernatant was discarded, the pellet was resuspended in 1 mL of PBS, and cells were counted. For every 1 × 10^7^ cells, 1 mL of TRIzol (Invitrogen, Waltham, MA, USA, catalog# 15596026) was used to resuspend the samples, which were then stored at −80 °C.

All centrifugation steps were performed at RT to isolate cells from the blood. Whole blood was centrifuged for 15 min at 1200× *g* and the buffy coat was collected and suspended in 15 mL of PBS. Using an underlay of Histopaque-1077 (Sigma-Aldrich, St. Louis, MO, USA, catalog #10771) for density centrifugation, cells were centrifuged at 1200× *g* for 15 min. The mononuclear cell portion was collected and suspended in 50 mL of PBS and centrifuged at 200× *g* for 20 min. The pellet was then washed 2 times using PBS and centrifuged at 400× *g* for 5 min. Any remaining red blood cells were lysed using sterile water and the blood mononuclear cell (BMC) portion was suspended in PBS and centrifuged at 400× *g* for 5 min. The BMCs were counted and cell viability was assessed using a Corning CytoSMART cell counter (Corning Life Sciences, Corning, NY, USA).

### 2.3. Magnetic-Activated Positive B Cell Sorting (MACS)

Cells isolated from blood underwent MACS. Colostral cells were not sorted since the B cell population is much smaller in colostrum and the high viscosity of colostrum may potentially reduce cell yield and affect the quality/integrity of the RNA.

Blood mononuclear cells (1 × 10^7^) were incubated at RT for 15 min with 250 µL of a 1/50 dilution of mouse IgG1 monoclonal anti-bovine IgM antibody (Sigma-Aldrich, catalog # I6137, clone BM-23, 4 µg/mL) and mouse IgG1 monoclonal anti-bovine IgG antibody (Sigma-Aldrich, catalog # B6901, clone BG-18, 5 µg/mL). Antibodies for labeling IgM+ and IgG+ B cells were included in the same reaction. Cells were washed twice with MACS buffer (PBS with 0.5% bovine serum albumin and 2 mM EDTA) and centrifuged for 5 min at 400× *g*. Next, cells were incubated in the dark at 4 °C for 15 min with rat anti-mouse IgG1 microbeads (20 μL of microbeads and 80 μL of MACS buffer for 10^7^ BMCs (Miltenyi Biotech, Bergisch Gladbach, Germany, catalog # 130-047-102)). Cells were then washed 2 times and sorted using a positive selection method. Cells were added to a MACS mini-column (Miltenyi Biotech, catalog # 130-042-201) per the manufacturer’s instructions. The positively (IgM+ and IgG+) sorted cells were collected, counted, and centrifuged for 5 min at 400 x g and, then, the supernatant was removed. Cells were resuspended and for every 1 × 10^7^ cells, they were stored in 1 mL of TRIzol (Invitrogen, Waltham, MA, USA, catalog# 15596026) at −80 °C.

### 2.4. cDNA and Polymerase Chain Reaction (PCR)

Samples from blood and colostrum were thawed on ice, 200 μL of chloroform was added, and samples were mixed by shaking. Blood and colostrum samples were processed using the same protocol for the following steps. Samples were centrifuged at 16,089× *g* for 15 min at 4 °C and, then, the aqueous phase containing RNA was collected. Next, 500 μL of isopropanol (≥99.5%) was added and the samples were vortexed and then incubated at −20 °C overnight. Samples were centrifuged at 16,089× *g* for 10 min at 4 °C, the supernatant was removed, and the RNA pellet was washed twice with 75% ethanol. The supernatant was removed and the RNA was resuspended in 12 μL of nuclease-free water. Samples were cleaned and concentrated using the MinElute kit (Qiagen, Venlo, NL, catalog#74204). An Agilent 4150 TapeStation System was used to ensure a RNA integrity number of ≥8 during optimization. The quality and quantity were checked using a DeNovix Spectrophotometer/Fluorometer (DS-11) to ensure a 260 nm/280 nm absorbance ratio of ≥1.8. Using 5000 ng of RNA and the SuperScript III First-Strand Synthesis System (Invitrogen, catalog # 18080051), cDNA was produced. In brief, the RNA, dNTP, diethylpyrocarbonate (DEPC)-treated water, and oligo (dT)20 were mixed and incubated at 65 °C for 5 min and then placed on ice. The reverse transcriptase buffer, MgCl2, DTT, RNase out, and SuperScript III reverse transcriptase were added and incubated for 50 min at 50 °C and the reaction was terminated at 85 °C for 5 min and placed on ice. RNase H was added and the tubes were incubated for 20 min at 37 °C. Samples were stored at −20 °C until they were used for downstream PCR applications.

The PCR reaction was prepared using the High-Fidelity PCR Master kit (Roche, Indianapolis, IN, USA, catalog # 12140314001) and adapted from the manufacturer’s protocol. First, 2 μL of cDNA was placed in a well of a Roche 96-well plate (catalog # 04729692001), then, 23 μL of mixed forward primer, reverse primer, and DEPC-treated water was added. Next, 25 μL of the high-fidelity master mix was added to each reaction. The forward primer, 5′-AGATGAACCCACTGTGGACC-3′, was used to amplify both IgM and IgG IGHV genes unbiasedly, adapted from a forward primer used by Saini et al. (1997) [27]. The reverse primer 5′-TGTTTGGGGCTGAAGTCC-3′ was used to amplify IgM heavy-chain regions, adapted from Ma et al. (2016) [31]. The same forward primer was used with reverse primer 5′-GCTGTGGTGGAGGCTGAG-3′ to amplify IgG heavy chains, adapted from Saini et al. (1997) [27]. The initial denaturation step was completed at 94 °C for 2 min for 1 cycle. For the amplification cycles, the denaturation step occurred at 94 °C for 15 s, annealing at 62 °C for 30 s, and the elongation step at 72 °C at 45 s for 10 cycles and then repeated for 10 more cycles while adding 5 s for each successive cycle of the elongation step. The final elongation step was completed at 72 °C for 7 min.

During the optimization, PCR products were evaluated by gel electrophoresis to assess if there was DNA contamination and whether PCR products were within the estimated base pair (bp) length. The PCR amplicons were cleaned and purified using the PureLink PCR Purification Kit (Invitrogen, catalog # K310001). Quantity and quality were checked to ensure a concentration of ≥20 ng/μL and a 260 nm/280 nm absorbance ratio ≥1.8 (DeNovix Spectrophotometer/Fluorometer (DS-11)) and samples were stored at −20 °C until sequencing was initiated.

### 2.5. MinION Sequencing

End preparation was completed by preparing 1 μg of the amplicon reaction with the NEBNext Ultra II End Repair/dA-Tailing Module (New England Biolabs, Ipswich, MA, USA, catalog # E7546S). Samples were washed and eluted in nuclease-free water. Samples were then barcoded and prepared using the Native barcoding kit 24 v14 (Oxford Nanopore Technologies, Oxford, UK, catalog # SQK-NBD114.24) and the Blunt/TA Ligase Master Mix (New England Biolabs, catalog # M0367S). Samples were washed and eluted in nuclease-free water. Adapters were added using the NEBNext Quick Ligation kit (New England Biolabs, catalog # E6056S) and then samples were washed and eluted in nuclease-free water. Colostrum and blood samples of both isotypes from the same animal were run on the same MinION Mk1C flow cell (Oxford Nanopore Technologies, catalog# R10.4.1) and allowed to run for ~72 h.

### 2.6. Cleaning

After sequencing concluded, basecalling and barcode sorting/trimming and adapter removal were completed using Oxford Nanopore Technology (ONT) Guppy basecaller software (v6.4.6 + ae70 e8f) (--flowcell FLO-MIN114,--kit SQK-LSK114, --detect_barcodes, --enable_trim_barcodes). FASTQ files containing the sequence ID, phred score, nucleotide (nt) sequence, and comments were retrieved for each barcoded sample. The quality of the sequences was checked using the FASTQC software (v0.11.9). The sequences were then filtered using the default parameters of fastp (v0.23.1) for quality (Q >= 15). Sequences were also filtered for length for IgM (--length_required 700 --length_limit 1200) or IgG (--length_required 200 --length_limit 800). Raw FASTQ files can be found in the NCBI sequence read archive under BioSample accession PRJNA1153617.

The minimum sequence length (700 nt) for IgM captures sequences from the forward primer (inclusive) to a well-defined 21 nt IgM motif from Walther et al. (2016) [32] corresponding to a very short-to-non-existent CDR3. The maximum length (1200 nt) captures sequences with the forward primer, the 21 nt IgM motif from Walther et al. (2016) [32], and may include the reverse primer with an ultralong CDR3 (defined as up to 225 nt between the pre- and post-CDR3 motifs, see motif definitions below).

For IgG sequences, the minimum sequence length (200 nt) allows the capture of sequences that contain a forward or reverse primer with a short CDR3. The maximum sequence length (800 nt) allows the capture of sequences that contain a forward primer to the reverse primer (IgG motif) and contain an ultralong CDR3, defined as up to 225 nt between the pre- and post-CDR3 motifs.

### 2.7. Filtering

Filtering the sequence motifs of interest was completed using seqkit (v2.3.1) and will be referred to as the refined stepwise literature-based (RSLB) filtering method. The RSLB filtering method was developed by consulting published Ig heavy-chain sequences and has been refined through optimization allowing minimal nt mismatches. The RSLB filtering method in this study was used to identify and provide CDR3 lengths by searching for primer sequences, bona fide isotype motifs, and sequences upstream and downstream of the CDR3.

#### 2.7.1. IgM Filtering Process

The first filtering step for IgM sequences was to search for the forward primer (AGATGAACCCACTGTGGACC) with the option of up to 4 nt mismatches. The allowance of options for nt mismatches at filtering steps was chosen after optimization for maximum capture of valid sequences. The file generated from the first step was then used to search for IgM motifs in the second step. The second step was to search for an isotype motif from Saini et al. (1999) [16] (AATCACACCCGAGAGTCTTC) with 4 nt mismatches or for a truncated version of an IgM motif from Walther et al. (2016) [32] (ACAGCCTCTCT) with the option of 2 nt mismatches (Figure 1).

A new file was generated for the third step, which searched the original filtered file for reverse orientation sequences that would not be picked up in the first or second steps. The third step was to search for the reverse IgM primer adapted from Ma et al. (2016) [31], (TGTTTGGGGCTGAAGTCC), with the option for 4 nt mismatches. All matched sequences from the third step were returned in the forward orientation. Finally, files from the second and third steps were merged. Duplicate sequences that were present in both files were removed to create a single file with only sequences in forward orientation.

#### 2.7.2. IgG Filtering Process

The first filtering step for IgG sequences was to search for the forward primer (AGATGAACCCACTGTGGACC) with the option of 4 nt mismatches. The file generated from the first step was then used to search for the IgG motif in the second step. The second step was to search for the IgG isotype motif from Saini et al. (1997) [27] (CTCAGCCTCCACCACAGC) with the option of up to 3 nt mismatches.

A new file was generated for the third step, which searched the original filtered file for reverse orientation sequences that would not be picked up in the first or second steps The third step was to search for the reverse IgG primer adapted from Saini et al. (1997) [27], (GCTGTGGTGGAGGCTGAG), with the option for 3 nt mismatches. All matched sequences from the third step were returned in the forward orientation. Finally, files from the second and third steps were merged. Duplicate sequences that were present in both files were removed to create a single file with only forward orientation sequences.

The pre-CDR3 was defined as the framework region 3 (FWR3) sequence immediately upstream from the CDR3, similar to the following sequence: “GAGGACACGGCCACATACTACTG”. A search for the pre-CDR3 was designed based on notarized bovine V genes obtained from the International ImMunoGeneTics Information System (IMGT (https://www.imgt.org/HighV-QUEST), accessed 6 July 2023 to 23 December 2023). After identifying sequences with a pre-CDR3, a search was carried out for post-CDR3 motifs immediately downstream of the CDR3 in FWR4, similar to “TGGGGCCAA”. The search motif was designed using notarized J genes from the IMGT in the FWR4 and common nucleotide variations in this area. Both pre- and post-CDR3 search motifs were adjusted to allow minor nucleotide substitutions commonly present in pre- and post-CDR3 sequences [33]. Regular expressions (regex) were used in seqkit to identify sequences with specific variations of nt patterns in the pre- and post-CDR3. A tsv file was generated containing DNA sequences that matched the pre- and post-CDR3 search; the file included sequence match, length of the matched sequence, and the sequence header. A Nextflow (v23.04.3) pipeline was implemented to complete all steps from FASTQC to generating the tsv file of filtered and matched DNA sequences. All scripts for the primary analysis and RSLB filtering can be found at: https://github.com/harohodg/filtering_bovineIgM_rep (accessed 1 January 2023 to 1 June 2024).

### 2.8. Sequence Analysis

After the CDR3s were identified, the aa length was calculated from C104 (in the V region) to W118 (beginning of the J region in FWR4), using the numbering system according to IMGT [34]. C104 and W118 were not included in the calculation for CDR3 length. CDR3s that were ≤3 nt or ≥225 nt were removed since these lengths would not correspond to a CDR3 of 1 to 75 aa, which was established as an arbitrary cut-off from the published literature. Sequences longer than 75 aa typically matched much further downstream (within the constant region) from the typical location of the post-CDR3 motif. The total number of sequences and the number of sequences with ultralong CDR3s (≥40 aa) were counted. The total number of sequences with an identified CDR3 was used as the denominator to estimate the percentage of sequences with ultralong CDR3s.

Productive sequences were also analyzed. First, using the filtering steps described above to isolate the CDR3 through the Nextflow RSLB pipeline, the sequences were translated from aa position 99 in FWR3 (upstream from CDR3) to 120 in FWR4 (downstream from CDR3). Stop codons were removed, and a tsv file was generated. Sequences with ≥40 aa from C104 to W118 were considered ultralong CDR3. Sequences with ≤10 aa were considered short CDR3s.

The IMGT HighV-Quest (v3.6.0) software was used to validate the filtering and ultralong CDR3 estimations [35]. IMGT HighV-Quest provided VDJ labeling, gene usage, and CDR3 length estimation for total sequences and per clonotype [35].

The Proc Corr function in SAS (Statistical Analysis Software, SAS OnDemand for Academics, 2024) was used to assess if there was a correlation in the percentage of IgM or IgG ultralong CDR3s between blood and colostrum or between isotypes in the blood and colostrum. A nonparametric Wilcoxon signed-rank test was used to compare the distribution of IgM or IgG ultralong CDR3 percentages and gene usage between the blood and colostrum. Significant differences were reported at *p* ≤ 0.05 and tendencies were reported for *p* > 0.05 to <0.10.

## 3. Results

### 3.1. Blood-Derived IgM Sequences with Ultralong CDR3s and VDJ Gene Usage

All estimates of the percentages of sequences coding for ultralong CDR3s and the number of sequences can be found in Table 1. The mean number of blood IgM raw read sequences was 384,776, ranging from 157,862 to 696,430 in seven Holstein cows. After the RSLB filtering method, 71,509 to 341,700 sequences were available for analysis. Among IgM DNA sequences, using the RSLB filtering method, the mean percentage of sequences with ultralong CDR3s (≥40 aa in length) was 6.46% (±1.11 standard error of the mean (SEM)) (Table 1). Using the RSLB filtering method, the mean percentage of productive IgM protein sequences with ultralong CDR3s was 4.22% (±0.70, Table 1, Figure 2).

IMGT was used to provide and validate the CDR3 length of total sequences and clonotypes and provide VDJ gene usage. In total sequences, the mean percent of ultralong CDR3s was 4.79% (±0.82) and on the basis of clonotypes was 5.08% (±0.86) (Table 1). On average, the percentage of sequences with ultralong CDR3s was 1.68% percentage points higher when using the RSLB filtering method compared to the IMGT total sequence analysis and 1.39% percentage points higher compared to the IMGT clonotypic analysis.

Using IMGT High-V/Quest analysis, the blood IgM V genes used most frequently are presented in Table 2 by clonotype and graphically presented in Figure 3. The usage of IGHV1-7 was of interest due to its reported high use in ultralong CDR3s [16]. The D genes most frequently used are listed in descending order: IGHD6-2 (18.50%), IGHD3-1 (14.64%), IGHD4-1 (14.24%), IGHD5-2 (12.18%), IGHD7-3 (10.10%), IGHD8-2 (9.34%), and IGHD6-3 (7.23%) (Figure 4). The order of D gene usage did not change between analyses based on total sequences and clonotypes. The usage of IGHD8-2 was of particular interest due to its reported high use in ultralong CDR3s [36]. There was high usage of IGHJ2-4 (97.37%), with no other genes used >1%.

### 3.2. Blood-Derived IgG Sequences with Ultralong CDR3s and VDJ Gene Usage

All estimates of the percentages of sequences coding for ultralong CDR3s and the number of sequences can be found in Table 3. The mean number of blood IgG raw read sequences was 685,851, ranging from 149,760 to 1,209,166 in seven Holstein cows. After the RSLB filtering method, 35,295 to 504,873 sequences were available for analysis. Among IgG DNA sequences, using the RSLB filtering method, the mean percentage of sequences with ultralong CDR3s was 9.72% (±1.47 SEM). Using the RSLB filtering method, the mean percentage of productive IgG protein sequences with ultralong CDR3s was 6.61% (±1.11, Table 3, Figure 2).

Using IMGT, in total blood IgG sequences, the mean ultralong CDR3s estimate was 7.22% (±1.02) and on the basis of clonotype was 10.93% (±0.91, Table 3). On average, the percentage of sequences with ultralong CDR3s was 2.50% percentage points higher when using the RSLB filtering method compared to the IMGT total sequence analysis. On average, the percentage of sequences with ultralong CDR3s was 1.21% percentage points higher when using IMGT clonotypic analysis compared to the RSLB filtering method.

Using the IMGT High-V/Quest analysis, V gene usage from blood IgG sequences is presented in Table 2 by clonotype and graphically presented in Figure 3. There was significantly greater usage of IGHV1-7 in blood IgG (16.78% ± 0.41) than in blood IgM sequences on a clonotypic basis (7.46% ± 0.27, *p* = 0.02). The following D genes were used in descending order and were similar for analyses based on total sequences and clonotype: IGHD6-2 (21.31%), IGHD4-1 (15.12%), IGHD3-1 (11.97%), IGHD7-3 (11.22%), IGHD8-2 (9.27%), IGHD5-2 (9.08%), and IGHD6-3 (7.92%, Figure 4). There was high usage of IGHJ2-4 (95.80%) with lower usage of IGHJ1-6 (2.14%) and IGHJ1-5 (1.24%) with no other J gene usage >1%.

### 3.3. Colostrum-Derived IgM Sequences with Ultralong CDR3s and VDJ Gene Usage

All estimates of the percentages of sequences coding for ultralong CDR3s and the number of sequences can be found in Table 4. The mean number of colostral IgM raw read sequences was 239,623, ranging from 132,933 to 365,677 in seven Holstein cows. After the RSLB filtering method, 61,647 to 165,151 sequences were available for analysis. Among colostral IgM DNA sequences using the RSLB filtering method, the mean estimate of ultralong CDR3s was 12.00% (±2.49 SEM). The mean estimate of productive IgM protein sequences that had ultralong CDR3s was 8.09% (±1.73). Using a Wilcoxon signed-rank test, there was a significantly greater percentage of productive IgM sequences with ultralong CDR3s in colostrum (8.09% ± 1.73) than in blood (4.22% ± 0.70, *p* = 0.048, Figure 2).

Using the IMGT High-V/Quest analysis, V gene usage in colostral IgM sequences is presented in Table 2 by clonotype and in Figure 3. There was significantly greater usage of IGHV1-7 in colostral IgM sequences (19.61% ± 2.76) than blood IgM sequences on a clonotypic basis (7.46% ± 0.77, *p* = 0.02). There was high usage of seven D genes described in descending order, in the same order based on total sequences, and by clonotype: IGHD6-2 (18.32%), IGHD4-1 (14.16%), IGHD8-2 (13.15%), IGHD3-1 (12.68%), IGHD7-3 (11.37%), IGHD5-2 (10.43%), and IGHD6-3 (8.71%, Figure 4). There was high usage of IGHJ2-4 (94.55%) with lower usage of IGHJ1-6 (1.61%) and IGHJ1-5 (1.58%) and no other J gene usage >1% (data not shown).

### 3.4. Colostrum-Derived IgG Sequences with Ultralong CDR3s and VDJ Gene Usage

All estimates of the percentages of sequences coding for ultralong CDR3s and the number of sequences can be found in Table 5. The mean number of colostral IgG raw read sequences was 516,975, ranging from 171,832 to 962,730 in seven Holstein cows (Table 5). After the RSLB filtering method, 61,504 to 371,226 sequences were available for analysis. Among colostral IgG DNA sequences, using the RSLB filtering method, the mean estimate of ultralong CDR3 sequences was 18.26% (±2.58 SEM, Table 5). The mean estimate of productive colostral IgG protein sequences with ultralong CDR3s was 12.98% (±1.98, Table 5). Using a Wilcoxon signed-rank test, there was a significantly greater percentage of IgG ultralong CDR3s in colostrum (12.98% ± 1.98) than in blood (6.61% ± 1.11, *p* = 0.048, Figure 2). The difference between the mean percent of IgG and IgM ultralong CDR3 sequences in colostrum was not significant (*p* = 0.58).

Using the IMGT analysis for colostral IgG DNA sequences, in total sequences, the mean estimate of ultralong CDR3s was 14.44% (±2.00) and on a clonotype basis was 23.02% (±2.79, Table 5). On average, the percent of sequences with ultralong CDR3s was 3.82% percentage points higher when using the RSLB filtering method compared to the IMGT total sequence analysis. In contrast, the mean percent of sequences with ultralong CDR3s was 4.76% percentage points higher when using IMGT clonotypic analysis compared to the RSLB filtering method.

Using the IMGT High-V/Quest analysis, the V genes most frequently used are presented in Table 2 by clonotype and graphically in Figure 3. The percentage usage of IGHV1-7 was significantly higher in colostral IgG sequences (27.46% ± 2.90) than in blood IgG sequences (16.78% ± 1.16, *p* = 0.03). The difference between the usage of IGHV1-7 in colostral IgG sequences (27.46% ± 2.90) and colostral IgM sequences was not significant (19.61% ± 2.76, *p* = 0.16). The following D genes were used in total sequences and by clonotype in descending order: IGHD6-2 (21.62%), IGHD8-2 (14.79%), IGHD4-1 (13.12%), IGHD7-3 (12.82%), IGHD3-1 (10.21%), IGHD5-2 (8.26%), and IGHD6-3 (6.77%) (Figure 4). There was high usage of the J gene IGHJ2-4 (95.11%), with no other usage except IGHJ1-6 (1.61) and IGHJ1-5 (1.41%) with >1% of sequences.

### 3.5. Blood- and Colostrum-Derived IgM and IgG Sequences with Short CDR3s

Data presented for blood- and colostrum-derived IgM and IgG sequences with short CDR3s are presented as RSLB-filtered productive protein sequences. The mean percentage of productive blood IgM sequences with short CDR3s (≤10 aa) was 3.48% (±0.44, Table 6). Using the RSLB filtering method, the mean percentage of productive blood IgG sequences with short CDR3s was 6.70% (±1.14, Table 6). Using the RSLB filtering method, the mean percentage of productive colostral IgM sequences with short CDR3s was 6.36% (±1.59, Table 6). Using the RSLB filtering method, the mean percentage of productive colostral IgG sequences with short CDR3s was 7.34% (±1.72 SEM, Table 6).

### 3.6. Summary of Statistical Findings

Between colostrum and blood, significant differences in the percent of ultralong CDR3s and VDJ gene usage were identified. Using a Wilcoxon signed-rank test, there was a significantly greater percentage of IgM productive protein sequences (by RSLB) with ultralong CDR3s in colostrum (8.09%, Table 6) than in blood (4.22%, *p* = 0.048, Table 1). There was significantly greater usage of IGHV1-7 in colostral IgM sequences (19.61%) than in blood IgM sequences on a clonotypic basis (7.46%, *p* = 0.02, Table 2).

Using a Wilcoxon signed-rank test, there was a significantly greater percentage of IgG ultralong CDR3s in colostrum (12.98%, Table 5) than in blood (6.61%, *p* = 0.048, Table 3). The percentage usage of IGHV1-7 was significantly greater in colostral IgG sequences (27.46%) than in blood IgG sequences (16.78%, *p* = 0.03). There was significantly greater usage of IGHV1-7 in blood IgG (16.78%) than in blood IgM sequences on a clonotypic basis (7.46%, *p* = 0.02).

### 3.7. Correlations

There was a tendency for a positive correlation between the percentage of IgM B cells with ultralong CDR3s in colostrum and in the blood for productive protein sequences (r = 0.69, *p* = 0.09, Figure 5). The correlation between the percentage of productive protein colostral IgG and blood IgG B cell sequences with ultralong CDR3s was not significant (r = −0.11, *p* = 0.81).

## 4. Discussion

In this study, colostral IgM and IgG B cells that express ultralong CDR3s were identified. In colostrum and blood, the percentages of ultralong CDR3s were greater in IgG sequences than in IgM sequences but the differences were not significant. It is possible that for IgG, affinity maturation (preferential selection), isotype switching, or somatic hypermutation contribute to the increase in the percentage of ultralong CDR3s. As hypothesized, in colostrum, both IgM and IgG sequences had a significantly greater percentage of ultralong CDR3s than samples from blood for the corresponding isotype.

In the published literature, the proportion of ultralong CDR3s in blood is estimated to range from 0.036% to 17.02% [16,33,37]. These estimates vary broadly, likely due to Ig isotype, age, breed, and laboratory methods. Among individual cows within this study, there is a large variation in the percent of ultralong CDR3 sequences. The large variation is especially apparent in the colostrum samples, which is reflected by the increased SEM among cows. It should be noted that, on average, there was a lower percentage of IgM B cells with ultralong CDR3s at the time of calving in this study compared to heifers and pregnant heifers that were 1 to 2 years of age, raised at the same facility [33]. To the authors’ knowledge, this is the first estimate of ultralong CDR3s at the time of calving and it would be useful to complete a longitudinal study throughout the reproductive cycle to solve this discrepancy. There are currently no estimates of the percentage of colostral B cells expressing ultralong CDR3s but the estimates from this study were higher than what is currently found in the literature regarding the percentage of B cells in blood with ultralong CDR3s. There are currently gaps in how age and life stage influence the BCR repertoire since most bovine BCR sequences are from fetuses [38], calves 1 to 2 months of age [37], calves approximately 5 months of age [39], and animals over 1 year of age [17]. The results from the current study provide insight into the percentage of sequences with ultralong CDR3s at the time of calving.

Laboratory methods, such as PCR and sequencing, may be biased against the detection of ultralong CDR3s. Shorter Ig heavy chains may be amplified preferentially, skewing the estimates to provide lower percentages of ultralong CDR3s. For that reason, investigating clonality may be valuable to provide more accurate estimates of the percentage of B cells with ultralong CDR3s. However, using sequencing platforms that have greater sequencing errors, can influence the designation of a clonotype. Clonality is determined by a unique VDJ rearrangement and CDR3 nucleotide junction sequence [35]. The potential for increased sequencing errors on the ONT platform may suggest there is more diversity in B cell populations than what is occurring in the repertoire. Additionally, low-throughput sequencing methods, such as earlier versions of long-read technology and Sanger sequencing, can decrease the likelihood of capturing rarer cell subsets, such as ultralong CDR3s. Older short-read technology may have difficulties covering the length of transcripts required without appropriate laboratory measures. Oyola et al. 2021 [17] reported that using nested PCR helped identify ultralong CDR3s that otherwise would have been overlooked. Finally, in this study, the RSLB filtering method typically provided a higher estimate of ultralong CDR3s than IMGT total sequence analysis but the highest measure of this difference was 3.82% percentage points. On a clonotype basis, IMGT provided higher estimates than the RSLB filtering method, with the highest measure of this difference being 4.76% percentage points. Previously, the RSLB filtering method and IMGT method were compared by the percentage of sequences with ultralong CDR3s [33]. On a DNA basis, the difference between filtering methods resulted in a 0.95 to 4.58 percentage point difference but the two methods were positively and significantly correlated (r= 0.99, *p* < 0.01) [33]. On a protein level, the difference ranged from 0.6 to 1.9 percentage points, except for one sample, which IMGT estimated higher by 4.32 percentage points. When translating sequences from RSLB-filtered DNA to productive protein, on average, there was a loss of 21.20% of the sequences, which is similar to sequence loss in previous studies [17,33,37].

The percentage of short CDR3s (≤10 aa) was similar for blood IgG (6.14%), blood IgM (3.25%), colostral IgM (6.81%), and colostral IgG (7.53%). Short CDR3s potentially allow more structural flexibility and play a role in stability to support the CDR1 and CDR2 regions for antigen binding. It has been reported that in humans and mice, there are specific B cells with short CDR3s that are particularly effective in neutralizing pathogenic viruses [40,41]. The role of antibodies with short CDR3s in cattle has not been investigated but future studies should investigate their structural and functional properties.

The usage of VDJ gene segments was compared in blood and in colostral IgM and IgG B cells. In colostrum on a clonotype basis, there was significantly greater usage of the *IGHV1-7* gene (highly used in the production of ultralong CDR3s), for both isotype IgG and IgM, compared to blood. The production of ultralong CDR3s in cattle is typically attributed to the usage of *IGHV1-7*, *IGHD8-2*, and *IGHJ2-4* [16,17,28,36]. It has been reported that >90% of sequences with ultralong CDR3s use *IGHV1-7* [16], 45% use *IGHD8-2* [33], and >95% use *IGHJ2-4* [17,33]. The high usage of *IGHV1-7* reflected the high percentages of sequences with ultralong CDR3s in colostrum. In blood, there was greater usage of genes such as *IGHV1-10*, *IGHV1-14*, and *IGHV1-7*. These findings suggest a greater usage of the *IGHV1-7* gene in colostral B cells, especially IgG B cells. The relative D gene usage in B cells from blood and colostrum was consistent; although, *IGHD8-2* was the second-highest-used gene in colostral IgG sequences whereas, in colostral IgM and blood IgM and IgG, the usage and rank were lower. The J gene usage across blood, colostrum, and isotypes remained consistent and is supported by previous estimates in the literature of 95% to 97% usage [17,33]. It is possible that B cells are able to move from circulation to the mammary gland and undergo clonal expansion events, which enrich certain gene use, or there may be a subset of B cells that are residing in the mammary gland.

The function and biological importance of B cells in colostrum in general, and B cells expressing ultralong CDR3s in colostrum in particular, is unknown. In humans and mice, milk-derived B cells have a unique profile. The reported phenotype is typically activated memory B cells with mucosal adhesion receptors [42]. The B cells infiltrate the mammary gland during late pregnancy and then are transferred through the milk into the neonate’s circulation [42]. Colostral B cells from the dam may be able to become activated to their cognate antigen in the calf and provide pathogen-specific responses [43] leading to maternal clones that can secrete antibodies. Colostral B cells may also help promote immune homeostasis in the neonate, which has been reported in mice studies with milk-derived IgG-producing cells [42]. Calves fed maternal colostral cells were seen to have improved immune responses measured by a mixed leukocyte response compared to calves fed cell-free colostrum from 24 h to 5 weeks of age [10]. It was hypothesized that colostral leukocytes play an important role in stimulating neonatal antigen-presenting cells and aid in protection against pathogens [10].

The destination and the life expectancy of colostrum-derived maternal cells are unknown but they may eventually reside in the lymphoid organs of the calf, such as the spleen, lymph nodes, Peyer’s patches, or the bone marrow. Reber et al. 2006 [44] fed fluorescently labeled (PKH26-GL) maternal cells to calves and maternal cells were detected in the neonates’ circulation at 12 h, peaked at 24 h, and disappeared by 36 h. The authors hypothesized that maternal cells may traffic to tissues and lymph nodes [44]. Langel et al. (2015) [2] reported an increase in the percentage of blood-derived CD4+ T cell subsets but not in relative B cell populations in calves fed whole colostrum versus cell-free colostrum. However, in a study investigating the effect of vaccination, calves fed cell-free colostrum had fewer relative numbers of B cells than calves fed whole colostrum [11]. It is unclear whether there is a shift in the relative B cell population in the first month of life depending on the colostrum source. Further investigation as to whether maternal B cells are influencing the repertoire of the neonate early in life is necessary. During pregnancy, fetal–maternal micro-chimerism occurs, where the bidirectional transfer and survival of fetal or maternal cells can occur in the fetus or mother [45]. Transfer of these cells promotes tolerance but may also have more implications and long-term effects, potentially on cells of the immune system [45].

Colostrum contains many cellular and acellular components that may work together to influence the immune response in the neonatal calf. Maternal B cells could play a role in activating cells, such as T cells, through antigen presentation. Components such as microRNAs, cytokines, Fc receptors, exosomes, and growth factors could all potentially have a direct influence on B cell function upon ingestion by the calf. Colostrum and milk contain miRNAs packaged in exosomes [46]. For example, miRNAs are often packaged in exosomes, which influence the development of the gut epithelial lining in the calf, aiding in cell proliferation [47]. MicroRNAs can also regulate immune responses and processes, such as B cell differentiation [47]. Additionally, the expression of different inhibitory and activating Fc receptors that bind the Fc region of maternal antibodies after colostrum consumption may generally influence the B cell response in the neonate [48].

Colostrum is a highly concentrated source of Igs but understanding how the bovine colostral BCR repertoire translates to secreted Ig protein profiles has yet to be elucidated. Studies in humans investigating the repertoire of circulating B cells in blood suggest there is only a limited correlation with the plasma antibody repertoire [49]. For this reason, it cannot be assumed that the B cell profile in colostrum reflects the secreted antibody profile in colostrum. In the future, studies that can estimate how well the B cell repertoire represents the secreted antibody profile in colostrum will be valuable. Additionally, these studies may assist in determining whether some of the antibodies in colostrum are being produced locally in the mammary gland or if they are largely derived from the dam’s circulation.

Bovine antibodies derived from blood and colostrum are of interest for treatment and therapeutic use in cattle and humans. Recently, researchers have been investigating the use of oral colostrum treatment in calves 2 to 21 days of age to help decrease morbidity in calves [50]. Although there is minimal absorption of Ig from the gut into circulation at an older age, there is a direct function at the gut level to control pathogens. Colostrum products can also be tailored towards specific pathogens of humans, such as hyperimmune bovine colostrum powder for travelers’ diarrhea [51]. Antibodies derived from blood from cattle with ultralong CDR3s are also being investigated on a therapeutic level for the neutralization of human-specific viral pathogens [29,52]. Antibodies are also being engineered to contain certain structures resembling the ultralong CDR3 knob and stalk domain [30]. These antibodies are promising as potential therapeutics and, at the very least, they provide a new avenue in antibody engineering and design [30]. Eventually, the enhancement of colostrum with antigen-specific bovine ultralong CDR3s for potential calf and human therapeutics may be possible.

One of the limitations of this study was the sample size. There were only seven colostrum samples and seven blood samples collected from cows. However, sequencing both IgM and IgG allowed a better understanding of the Ig repertoire and eventually can be applied to all isotypes. Most sequencing studies investigating ultralong CDR3s have had small sample sizes, typically less than four cows [16,17,37]. However, one study has used short-read sequencing to investigate the repertoire responses of 204 purebred Black Angus calves following vaccination [39]. In the current study, among the seven cows, there was a large variation among individual animals in the percentage of sequences with ultralong CDR3s and in VDJ gene usage. This variation must be explored further to understand the impact of parity, the volume of colostrum produced, genetics, and health status.

Another limitation of this study is the low number of B cells in colostrum, specifically IgM B cells. For this reason, cells from colostrum did not undergo MACS because of the potentially small quantity of B cells present in colostral secretions paired with the difficulty associated with this sample type. It is unknown whether the use of MACS, per se, contributed to the observed difference in the percent of ultralong CDR3 sequences in the current study. Ultimately, PCR was necessary to provide amplicons from the total cell fraction of colostrum. The small number of IgM B cells in colostrum motivated the use of PCR. However, it is anticipated that PCR bias would be consistent across all samples [39] and was needed to facilitate this first investigation into the VDJ gene usage by B cells in colostrum. Future studies plan to investigate if there are shared BCR clonotypes and patterns of VDJ gene usage and somatic mutations between the colostrum from the dam and the circulation of the calf before and after colostrum consumption. With newer technologies, analyzing B cell repertoires without the need to complete PCR will strengthen the foundational knowledge of the repertoire.

## 5. Conclusions

The current study identified and provided an estimate of the percentage of ultralong CDR3s and VDJ gene usage in colostral IgM and IgG B cell repertoires while contrasting with those of B cells in the blood. As hypothesized, higher percentages of IgM and IgG B cells in colostrum expressed ultralong CDR3s than the corresponding isotypes in the blood. There was greater usage of the *IGHV1-7* gene in B cells from colostral samples than from blood for the corresponding isotypes. The influence and fate of maternal B cells in colostrum that are potentially transferred to the neonatal calf are unclear. Characterization of the populations of B cells present in the colostrum will be the first step to decipher their influences on B cell development and immune protection in the neonatal calf. Detection of the homing sites of the maternal B cells within the calf and their persistence over time will also be critical for understanding functions. Additionally, it would be informative to determine whether these B cells are derived from the circulation or from B cells resident in mammary tissue.

## Figures and Tables

**Figure 1 animals-14-02841-f001:**
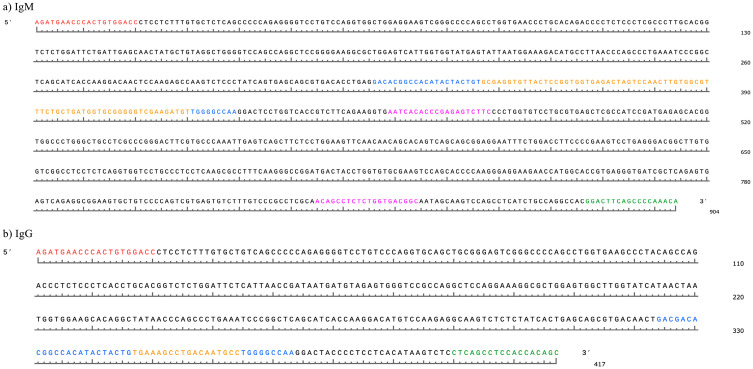
Visualization of filtering steps for IgM and IgG sequences derived from the blood sample of Cow 1. (**a**) The first step for IgM sequence identification was to search for the forward primer, which is in bright red font. The second step was to search the file generated from the first step (forward primer+) for two IgM isotype motifs, which are noted in magenta font. The third step, using the original file, was to search for the reverse primer (in reverse complement orientation) to identify sequences in reverse complement orientation, which is noted in the green font (it occurs in this image in forward orientation since this example is a forward-oriented sequence). The final step was to search for the pre- and post-CDR3 motifs, which are in blue font. (**b**) The first step for IgG sequence identification was to search for the forward primer, which is in bright red font. The second step was to search the file generated from the first step (forward primer+) for the IgG isotype motif, which is noted in green font. The third step, using the original file, was to search for the reverse primer (in reverse complement orientation) to identify sequences in reverse complement orientation (the reverse primer in this example is in green font, since it is the same motif from the second step, but was searched in reverse complement orientation). The final step was to search for the pre- and post-CDR3 motifs, which are in blue font.

**Figure 2 animals-14-02841-f002:**
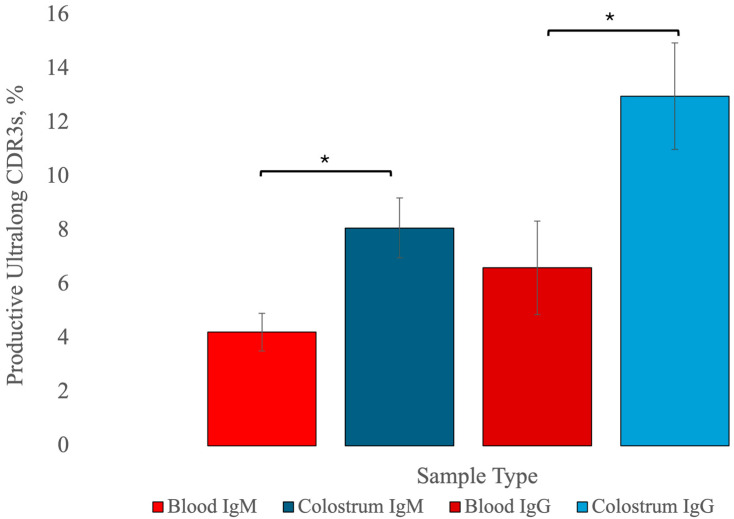
The percentage of IgM and IgG sequences that contain an ultralong complementarity determining region 3 (CDR3). In colostrum, there was a significantly greater percentage of IgM B cells with ultralong CDR3s compared to blood (*p* = 0.048). There was a significantly greater percentage of IgG B cells in colostrum with ultralong CDR3s compared to blood (*p* = 0.048). Significant differences of p <0.05 are depicted by *.

**Figure 3 animals-14-02841-f003:**
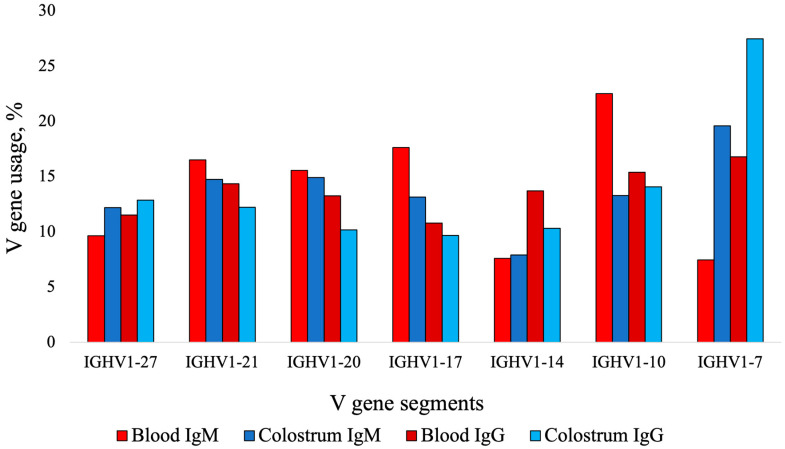
Percent V gene usage in colostrum and blood IgM and IgG sequences on a clonotype basis. The top seven used V genes are described for each sample type and isotype. Gene usage was allocated using the ImMunoGeneTics Information System (IMGT) based on DNA sequences. There is preferential use of IGHV1-7 in ultralong CDR3 sequences.

**Figure 4 animals-14-02841-f004:**
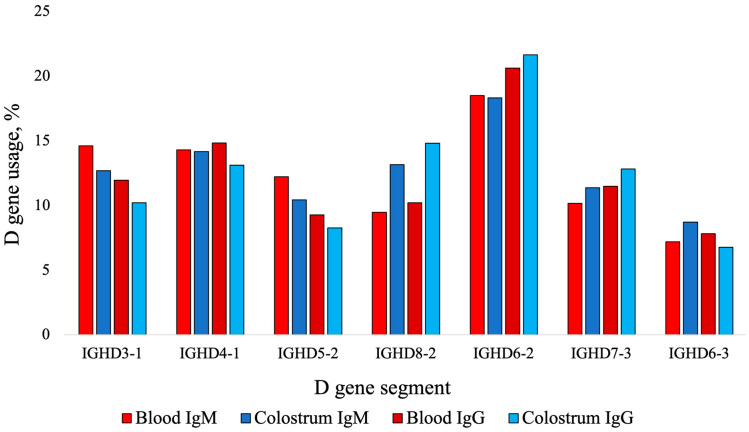
Percent D gene usage in colostrum and blood IgM and IgG sequences on a clonotype basis. The top seven used D genes are described for each sample type and isotype. Gene usage was allocated using the ImMunoGeneTics Information System (IMGT) based on DNA sequences. There is the preferential use of IGHD8-2 in ultralong CDR3 sequences.

**Figure 5 animals-14-02841-f005:**
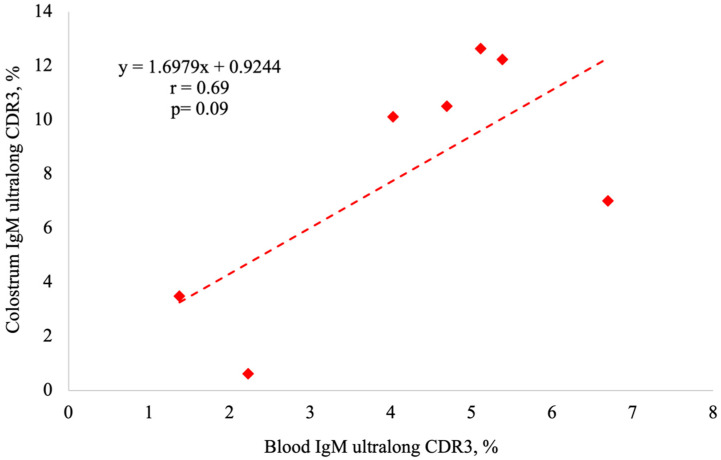
Percentage of RSLB-filtered productive protein IgM sequences with ultralong CDR3s in colostrum versus the percentage of RSLB-filtered productive protein IgM sequences with ultralong CDR3s in blood. Using the Proc Corr function in SAS, there was a tendency for a positive correlation between the percentage of ultralong CDR3 sequences in colostrum and blood (*p* < 0.09).

**Table 1 animals-14-02841-t001:** Blood IgM sequences with ultralong CDR3s estimated using RSLB filtering steps and IMGT resources with DNA sequences and productive protein sequences.

	# of Raw Reads	DNA RSLB Ultralong CDR3, %	IMGT Total Ultralong CDR3, %	IMGT Clonotype Ultralong CDR3, %	Productive Protein RSLB Ultralong CDR3, %	Parity
**Cow 1**		**3.56 ^1^**	**2.70**	**2.83**	**2.23**	3
# of sequences	497,638	215,330 ^2^	100,082	130,754	173,940	
**Cow 2**		**5.86**	**4.46**	**4.69**	**4.03**	3
# of sequences	157,862	71,509	46,142	43,584	57,774	
**Cow 3**		**8.20**	**5.99**	**6.17**	**5.39**	2
# of sequences	436,718	185,161	118,255	114,368	145,390	
**Cow 4**		**10.84**	**8.18**	**8.58**	**6.70**	1
# of sequences	255,239	101,786	57,033	54,193	75,862	
**Cow 5**		**7.69**	**5.44**	**5.95**	**5.12**	1
# of sequences	317,908	139,049	91,368	80,896	110,377	
**Cow 6**		**6.96**	**5.16**	**5.61**	**4.70**	1
# of sequences	331,635	157,811	105,500	96,091	128,226	
**Cow 7**		**2.14**	**1.59**	**1.70**	**1.38**	1
# of sequences	696,430	341,700	219,045	204,463	277,010	
**Mean**	384,776	**6.46**	**4.79**	**5.08**	**4.22**	
**SEM**		1.11	0.82	0.86	0.70	

^1^ The percentage of ultralong complementarity determining region 3 (CDR3) from sequencing the bovine blood IgM B cell repertoire. The estimates of the percentage of sequences with ultralong CDR3s were generated through the refined stepwise literature-based (RSLB) filtering method. The ImMunoGeneTics Information System (IMGT) program generated estimates of ultralong CDR3s as total sequences and on a clonotype basis. The percentage of productive protein sequences with ultralong CDR3s was generated by using the RSLB filtering method and then translating sequences from the pre-CDR3 to post-CDR3 and removing sequences with stop codons. ^2^ The number of DNA sequences generated from the RSLB filtering method or IMGT. The number of sequences is the total number of sequences that remained after each filtering method, which was the denominator used to generate the percentage of sequences with ultralong CDR3s.

**Table 2 animals-14-02841-t002:** V gene usage in blood and colostrum IgM and IgG sequences using the IMGT VDJ gene labeling by clonotype in Holstein cows.

	V Genes ^1^						
Sample	*IGHV1-27*	*IGHV1-21*	*IGHV1-20*	*IGHV1-17*	*IGHV1-14*	*IGHV1-10*	*IGHV1-7*
**Blood IgM**	9.65 ^1^	16.50	15.56	17.63	7.60	22.50	7.46
**Blood IgG**	11.53	14.35	13.27	10.80	13.69	15.38	16.78
**Colostrum IgM**	12.21	14.75	14.92	13.16	7.91	13.28	19.61
**Colostrum IgG**	12.86	12.22	10.18	9.67	10.32	14.08	27.46

^1^ The mean usage of the seven most highly used variable (V) genes. V gene usage was obtained on a clonotype basis from the ImMunoGeneTics Information System for each animal and for each sample type and then the mean usage of the gene was calculated by sample type (blood *n* = 7, colostrum *n* = 7) and isotype. The *IGHV1-7* gene is preferentially used in sequences with ultralong CDR3s.

**Table 3 animals-14-02841-t003:** Blood IgG sequences with ultralong CDR3s estimated using RSLB filtering steps and IMGT resources with DNA sequences and productive protein sequences.

	# of Raw Reads	DNA RSLB Ultralong CDR3, %	IMGT Total Ultralong CDR3, %	IMGT Clonotype Ultralong CDR3, %	Productive Protein RSLB Ultralong CDR3, %	Parity
**Cow 1**		**5.85 ^1^**	**4.57**	**8.09**	**3.88**	3
# of sequences	775,589	309,822 ^2^	221,948	100,082	251,431	
**Cow 2**		**14.97**	**10.41**	**13.38**	**10.86**	3
# of sequences	149,760	35,295	41,924	29,308	27,604	
**Cow 3**		**4.75**	**3.56**	**8.01**	**3.04**	2
# of sequences	930,963	399,527	282,485	94,177	320,304	
**Cow 4**		**7.58**	**5.80**	**9.17**	**4.66**	1
# of sequences	1,209,166	504,873	306,619	181,970	385,161	
**Cow 5**		**11.28**	**8.61**	**12.85**	**7.49**	1
# of sequences	704,778	274,667	201,418	122,482	215,610	
**Cow 6**		**9.88**	**7.34**	**12.20**	**6.75**	1
# of sequences	844,737	360,397	260,017	137,433	288,277	
**Cow 7**		**13.76**	**10.28**	**12.81**	**9.59**	1
# of sequences	185,961	56,664	53,575	38,914	44,510	
**Mean**	685,851	**9.72**	**7.22**	**10.93**	**6.61**	
**SEM**		1.47	1.02	0.91	1.11	

^1^ The percentage of ultralong complementarity determining region 3 (CDR3) from sequencing the bovine blood IgG B cell repertoire. The estimates of the percentage of sequences with ultralong CDR3s were generated through the refined stepwise literature-based (RSLB) filtering method. The ImMunoGeneTics Information System (IMGT) program generated estimates of ultralong CDR3s as total sequences and on a clonotype basis. The percentage of productive protein sequences with ultralong CDR3s was generated from the RSLB filtering method and then translating sequences from the pre-CDR3 to post-CDR3 and removing sequences with stop codons.^2^ The number of DNA sequences generated from the RSLB filtering method or IMGT. The number of sequences is the total number of sequences that remained after each filtering method, which was the denominator used to generate the percentage of sequences with ultralong CDR3s.

**Table 4 animals-14-02841-t004:** Colostral IgM sequences with ultralong CDR3s estimated using RSLB filtering steps and IMGT resources with DNA sequences and productive protein sequences.

	# of Raw Reads	DNA RSLB Ultralong CDR3, %	IMGT Total Ultralong CDR3, %	IMGT Clonotype Ultralong CDR3, %	Productive Protein RSLB Ultralong CDR3, %	Parity
**Cow 1**		**1.23 ^1^**	**1.14**	**4.38**	**0.62**	3
# of sequences	365,677	165,151 ^2^	127,307	22,600	137,907	
**Cow 2**		**14.36**	**10.10**	**19.57**	**10.11**	3
# of sequences	179,539	87,438	64,482	17,423	70,251	
**Cow 3**		**17.54**	**12.73**	**22.49**	**12.24**	2
# of sequences	132,933	61,647	41,372	17,307	48,237	
**Cow 4**		**10.87**	**8.16**	**15.18**	**7.01**	1
# of sequences	234,861	92,554	56,091	27,109	69,879	
**Cow 5**		**17.38**	**13.02**	**24.23**	**12.64**	1
# of sequences	257,848	115,503	81,483	30,824	91,572	
**Cow 6**		**17.51**	**11.21**	**20.09**	**10.51**	1
# of sequences	300,714	142,558	97,306	35,194	113,210	
**Cow 7**		**5.10**	**3.92**	**10.15**	**3.48**	1
# of sequences	205,788	109,198	79,278	19,393	92,796	
**Mean**	239,623	**12.00**	**8.61**	**16.58**	**8.09**	-
**SEM**		2.49	1.71	2.70	1.73	

^1^ The percentage of ultralong complementarity determining region 3 (CDR3) from sequencing the bovine colostral IgM B cell repertoire. The estimates of the percentage of sequences with ultralong CDR3s were generated through the refined stepwise literature-based (RSLB) filtering method. The ImMunoGeneTics Information System (IMGT) program generated estimates of ultralong CDR3s as total sequences and on a clonotype basis. The percentage of productive protein sequences with ultralong CDR3s was generated from the RSLB filtering method and then translating sequences from the pre-CDR3 to post-CDR3 and removing sequences with stop codons. ^2^ The number of DNA sequences generated from the RSLB filtering method or IMGT. The number of sequences is the total number of sequences that remained after each filtering method, which was the denominator used to generate the percentage of sequences with ultralong CDR3s.

**Table 5 animals-14-02841-t005:** Colostral IgG sequences with ultralong CDR3s estimated using RSLB filtering steps and IMGT resources with DNA sequences and productive protein sequences.

	# of Raw Reads	DNA RSLB Ultralong CDR3, %	IMGT Total Ultralong CDR3, %	IMGT Clonotype Ultralong CDR3, %	Productive Protein RSLB Ultralong CDR3, %	Parity
**Cow 1**		**19.85 ^1^**	**18.24**	**21.15**	**18.03**	3
# of sequences	511,813	169,945 ^2^	132,546	40,611	134,194	
**Cow 2**		**9.44**	**7.15**	**13.80**	**6.67**	3
# of sequences	244,094	94,780	74,925	27,425	73,284	
**Cow 3**		**16.79**	**13.62**	**19.84**	**10.96**	2
# of sequences	261,534	76,998	72,384	44,238	57,739	
**Cow 4**		**11.03**	**8.64**	**16.47**	**6.96**	1
# of sequences	962,730	370,121	252,572	113,456	278,054	
**Cow 5**		**17.25**	**13.17**	**24.74**	**11.89**	1
# of sequences	625,793	254,916	224,483	87,308	201,066	
**Cow 6**		**26.17**	**20.47**	**32.41**	**19.54**	1
# of sequences	841,032	371,226	274,073	91,643	282,998	
**Cow 7**		**27.26**	**19.77**	**32.71**	**16.79**	1
# of sequences	171,832	61,504	55,111	21,776	45,795	
**Mean**	**516,975**	**18.26**	**14.44**	**23.02**	**12.98**	
**SEM**		2.58	2.00	2.79	1.98	

^1^ The percentage of sequences with ultralong complementarity determining region 3 (CDR3) from sequencing the bovine colostral IgG B cell repertoire. The estimates of the percentage of sequences with ultralong CDR3s were generated through the refined stepwise literature-based (RSLB) filtering method. The ImMunoGeneTics Information System (IMGT) program generated estimates of ultralong CDR3s as total sequences and on a clonotype basis. The percentage of productive protein sequences with ultralong CDR3s was generated from the RSLB filtering method and then translating sequences from the pre-CDR3 to post-CDR3 and removing sequences with stop codons. ^2^ The number of DNA sequences generated from the RSLB filtering method or IMGT. The number of sequences is the total number of sequences that remained after each filtering method, which was the denominator used to generate the percentage of sequences with ultralong CDR3s.

**Table 6 animals-14-02841-t006:** Blood and colostral IgM and IgG short CDR3s percentages in productive protein sequences using the refined stepwise literature-based filtering method in Holstein cows at calving.

	Blood IgM Short CDR3s, % ^1^	Colostral IgM Short CDR3s, % ^2^	Blood IgG Short CDR3s, % ^1^	Colostrum IgG Short CDR3s, % ^2^	Parity
Cow 1	2.98	1.75	7.13	1.80	3
Cow 2	2.48	2.54	4.46	5.01	3
Cow 3	2.16	8.31	2.22	11.47	2
Cow 4	3.44	3.69	9.48	8.74	1
Cow 5	5.67	8.47	11.25	13.36	1
Cow 6	4.13	6.00	6.22	9.11	1
Cow 7	3.47	13.77	6.16	1.88	1
**Mean**	3.48	6.36	6.70	7.34	
**SEM**	0.44	1.59	1.14	1.72	

^1^ The percentage of sequences with short complementarity determining region 3 (CDR3) in B cells from blood. B cells were extracted using magnetic-activated cell sorting, RNA and cDNA were produced, and then PCR was used to amplify IgM and IgG heavy chains. Samples were then sequenced using Oxford Nanopore’s MinION with R10.4 Sequences with short CDR3s in the isotypes IgM and IgG were estimated using the refined stepwise literature-based filtering method to identify CDR3s and productive protein CDR3s. Sequences with ≤10 amino acids were considered to have a short CDR3. ^2^ The percentage of sequences with short complementarity determining region 3 (CDR3) in B cells from colostrum. Colostral cells were isolated, RNA was extracted, cDNA was produced, and then PCR was used to amplify IgM and IgG heavy chains. Samples were then sequenced using Oxford Nanopore’s MinION with R10.4. Short CDR3s in the isotypes IgM and IgG were estimated using the refined stepwise literature-based filtering method to identify CDR3s and productive protein CDR3s. Sequences with ≤10 amino acids were considered to have a short CDR3.

## Data Availability

Raw FASTQ files can be found in the NCBI sequence read archive under BioSample accession PRJNA1153617.

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
